# Harmonic-Gaussian Symmetric and Asymmetric Double Quantum Wells: Magnetic Field Effects

**DOI:** 10.3390/nano13050892

**Published:** 2023-02-27

**Authors:** Esin Kasapoglu, Melike Behiye Yücel, Carlos A. Duque

**Affiliations:** 1Physics Department, Science Faculty, Sivas Cumhuriyet University, Sivas 58140, Turkey; 2Physics Department, Science Faculty, Akdeniz University, Antalya 07058, Turkey; 3Grupo de Materia Condensada-UdeA, Instituto de Física, Facultad de Ciencias Exactas y Naturales, Universidad de Antioquia UdeA, Calle 70 No. 52-21, Medellín AA 1226, Colombia

**Keywords:** harmonic-Gaussian potential, double quantum well, magnetic field

## Abstract

In this study, we considered the linear and non-linear optical properties of an electron in both symmetrical and asymmetrical double quantum wells, which consist of the sum of an internal Gaussian barrier and a harmonic potential under an applied magnetic field. Calculations are in the effective mass and parabolic band approximations. We have used the diagonalization method to find eigenvalues and eigenfunctions of the electron confined within the symmetric and asymmetric double well formed by the sum of a parabolic and Gaussian potential. A two-level approach is used in the density matrix expansion to calculate the linear and third-order non-linear optical absorption and refractive index coefficients. The potential model proposed in this study is useful for simulating and manipulating the optical and electronic properties of symmetric and asymmetric double quantum heterostructures, such as double quantum wells and double quantum dots, with controllable coupling and subjected to externally applied magnetic fields.

## 1. Introduction

It is very important to construct a universal empirical potential energy function for diatomic and/or polyatomic molecules. For example, the first simple empirical analytical potential function proposed by Morse in 1929 [1] was used to study transition frequencies and intensities in a series of diatomic and polyatomic molecules [2]. For diatomic molecules, by employing the dissociation energy and the equilibrium bond length as explicit parameters, the Rosen–Morse, Manning–Rosen, Schiöberg, and Tietz potential-energy functions have been generated [3,4,5]. The modified Lennard–Jones potential energy function [6] has been used to perform potential fits experimental data to diatomic molecules.

As is known, double quantum wells (DQW) that characterize the bilayer systems are the semiconductor heterostructures exhibiting tunnel coupling. DQWs, which consist of various semiconductor materials, frequently appear in lasers emitting light in a wide range of wavelengths [7,8]. DQW’s potential energy functions, suggested to obtain information about diatomic molecules, are known as quasi-exactly solvable (QES) potentials. The quartic [9], sextic–decatic [10], Razavy [11], and Manning [12] double well potentials which provide a useful approximation for the potential energy of a diatomic molecule are some of them. Dong and Lemus reported the ladder operators for the modified Pöschl–Teller potential [13]. Particularly, they found a closed form of the normalization constants of the wave function by using two different methods and calculated analytical expressions for the matrix elements derived from the ladder operators. Using the exact quantization rule, Gu et al. calculated the energy spectra for modified Rosen–Morse potential [14]. In the same way, Dong et al. reported semi-exact solutions of the Razavy potential [15]. In their work, they show how to find the wave function exact solutions, which are given by the confluent Heun functions. Additionally, their method has been extended to the calculations of the asymmetric double well potential [16].

The optical properties of semiconductor quantum wells depend on the asymmetry of the confinement potential. So, the optical properties of the low dimensional heterostructures that are either with the inherent asymmetric character without an electric field or symmetrical character under an electric field have been studied intensively [17,18,19,20,21,22,23,24,25]. In this context, in this study, we examined the linear and non-linear optical properties of an electron in both symmetrical and asymmetrical DQW, which consist of the sum of an internal Gaussian barrier and a harmonic potential under an applied magnetic field.

To our knowledge, such a study has not yet been reported. This potential formed by the sum of a harmonic and Gaussian potential has been used to study the eigenstates in ammonia (umbrella inversion in ammonia) [26], and the proton transfer between two water molecules [27]. In these cases, the potential is symmetric, and the Gaussian maximum coincides with the parabola minimum. The inversion of ammonia, in which the hydrogen atoms pass from one side of the nitrogen atom to the other, is a significant problem that has been studied by many researchers. The potential function for the vibration leading to inversion is generally considered a harmonic potential with a potential barrier to hinder.

The harmonic oscillator (HO) potential is used to describe a molecular vibration in the very close neighborhood of a stable equilibrium point. This is one of the few quantum-mechanical systems with exact and analytical solutions. The Gaussian barrier within the dominant harmonic potential causes a bunching between adjacent symmetric (+) and anti-symmetric (−) states. All energy states increase by the presence of the barrier, but energies of anti-symmetric states increase less than the symmetric ones since asymmetric states have a node in the barrier that is not in symmetrical states.

The work is organized as follows: we describe the theoretical framework in Section 2. In Section 3, we discuss the obtained electronic and optical properties, and finally, the conclusions are found in Section 4.

## 2. Theoretical Model

In the effective mass approximation, the Hamiltonian for an electron under an applied magnetic field can be expressed as,
(1)$$H=\frac{1}{2m^{*}}{\left[\vec{p}+\frac{e}{c}\vec{A}\left(\vec{r}\right)\right]}^{2}+V\left(z\right)$$
or
(2)$$H=\frac{-\hbar^{2}}{2m^{*}}\frac{d^{2}}{dz^{2}}+\frac{e^{2}B^{2}z^{2}}{2m^{*}c^{2}}+V\left(z\right)\;,$$
where the magnetic field-$\vec{B}$ is applied perpendicular to the growth direction, $\vec{A}=\left(0,-B\;z,0\right)$ is the vector potential associated with the magnetic field, $\vec{p}$ is the momentum operator, $m^{*}$ is the electron effective mass, *e* is the elementary charge, and V(z) is the confinement potential of harmonic-Gaussian double quantum well (H-G DQW). Its functional form is given as follows [28]
(3)$$V\left(z\right)=V_{0}\;\left[A_{1}\;{\left(z/k\right)}^{2}+A_{2}\;e^{-{\left(\frac{z}{k}-z_{0}\right)}^{2}}\right]\;,$$
where $V_{0}$ is the depth of the quantum well, the *k*-parameter is related to the well and barrier width, $z_{0}$ is the asymmetry parameter, $A_{1}$ and $A_{2}$ are the structural parameters that adjust the coupling between the wells, well width and barrier height. For example, as the parameter $A_{1}$ increases, the well width becomes narrow. The $A_{2}$ parameter is related to the barrier height.

After the energies and related wave-functions are acquired, the linear and non-linear absorption coefficients are found using the perturbation expansion and the density matrix methods for transitions between two electronic states. The linear, third-order non-linear, and total absorption coefficients (TACs) are found as follows [21,28,29,30,31,32], respectively,
(4)$$\beta^{\left(1\right)}\left(\omega \right)=\sqrt{\frac{\mu_{0}}{\varepsilon_{R}}}\frac{|M_{ij}{|}^{2}\sigma_{\nu }\hbar \omega \Gamma_{ij}}{{\left(E_{ij}-\hbar \omega \right)}^{2}+{\left(\hbar \Gamma_{ij}\right)}^{2}}\;,$$
(5)$$\begin{array}{cc} \beta^{\left(3\right)}\left(\omega ,I\right)= & -2\sqrt{\frac{\mu_{0}}{\varepsilon_{R}}}\left(\frac{I}{\varepsilon_{0}n_{r}c}\right)\frac{|M_{ij}{|}^{4}\sigma_{\nu }\hbar \omega \Gamma_{ij}}{{\left[{\left(E_{ij}-\hbar \omega \right)}^{2}+{\left(\hbar \Gamma_{ij}\right)}^{2}\right]}^{2}}\;\\ \; & \times \left[1-\frac{|M_{jj}-M_{ii}{|}^{2}}{|2M_{ij}{|}^{2}}\frac{{\left(E_{ij}-\hbar \omega \right)}^{2}-{\left(\hbar \Gamma_{ij}\right)}^{2}+2E_{ij}\left(E_{ij}-\hbar \omega \right)}{E_{ij}^{2}+{\left(\hbar \Gamma_{ij}\right)}^{2}}\right]\;, \end{array}$$
and
(6)$$\beta \left(\omega \right)=\beta^{\left(1\right)}\left(\omega \right)+\beta^{\left(3\right)}\left(\omega \right)\;.$$

In the case of the relative changes of the refraction index coefficient, the corresponding expressions are
(7)$$\frac{\Delta n^{\left(1\right)}\left(\omega \right)}{n_{r}}=\frac{\sigma_{v}\;{\left|M_{ij}\right|}^{2}}{2\;\varepsilon_{0}\;n_{r}^{2}}\;\;\frac{E_{ij}-\hbar \;\omega }{{\left(E_{ij}-\hbar \;\omega \right)}^{2}+{\left(\hbar \;\Gamma_{ij}\right)}^{2}}\;,$$
(8)$$\begin{array}{c} \frac{\Delta n^{\left(3\right)}\left(\omega ,I\right)}{n_{r}}=-\frac{\mu_{0}\;c\;I\;\sigma_{v}\;{\left|M_{ij}\right|}^{2}}{4\;\varepsilon_{0}\;n_{r}^{3}}\;\;\frac{E_{ij}-\hbar \;\omega }{{\left[{\left(E_{ij}-\hbar \;\omega \right)}^{2}+{\left(\hbar \;\Gamma_{ij}\right)}^{2}\right]}^{2}}\;\\ \times \left[4\;|M_{ij}{|}^{2}-\frac{|M_{jj}-M_{ii}{|}^{2}}{E_{ij}^{2}+{\left(\hbar \;\Gamma_{ij}\right)}^{2}}\;\;\left\{E_{ij}\;\left(E_{ij}-\hbar \;\omega \right)-{\left(\hbar \;\Gamma_{ij}\right)}^{2}-{\left(\hbar \;\Gamma_{ij}\right)}^{2}\frac{\left(2\;E_{ij}-\hbar \;\omega \right)}{\left(E_{ij}-\hbar \;\omega \right)}\right\}\right]\;, \end{array}$$
and
(9)$$\frac{\Delta n(\omega ,I)}{n_{r}}=\frac{\Delta n^{\left(1\right)}\left(\omega \right)}{n_{r}}+\frac{\Delta n^{\left(3\right)}\left(\omega ,I\right)}{n_{r}}\;,$$
Here, $\varepsilon_{R}=n_{r}^{2}\;\varepsilon_{0}$ is the real part of the permittivity, $\varepsilon_{0}$ is the permittivity of vacuum, $n_{r}$ = $\sqrt{\varepsilon_{r}}$, is the refraction index, $\sigma_{\nu }$ is the carrier density in the system, $\mu_{0}$ is the vacuum permeability, $E_{ij}=E_{j}-E_{i}$ is the energy difference between two electron states, $M_{ij}=|\langle \psi_{i}\left|e\;z\right|\psi_{j}\rangle |$, ((i,j = 1,2)) is the dipole matrix element between the eigenstates $\psi_{i}$ and $\psi_{j}$ for incident radiation polarized in the *z*-direction, $\Gamma_{ij}=\left(1/T_{ij}\right)$ is the relaxation rate, $T_{ij}$ is the inverse relaxation time, *c* is the speed of the light in free space, and *I* is the intensity of incident photon with the ω-angular frequency that leads to the intersubband optical transitions. It should be noted that we will use the reduced dipole matrix element (RDME) definition ($\eta =M_{ij}/e$) in the length dimension in the figures.

## 3. Results and Discussion

To perform our numerical calculations the parameters are: $\varepsilon_{r}=12.58$, $m^{*}=0.067\;m_{0}$ (where $m_{0}$ is the free electron mass), $V_{0}=228$ meV, $T_{ij}=0.2$ ps, $\mu_{0}=4\pi \times 10^{-7}$ H m${}^{-1}$, $\sigma_{\nu }=3.0\times 10^{22}$ m${}^{-3}$, and $I=5.0\times 10^{8}$ W/m${}^{2}$. The value used for the width parameter in this study is k=20 nm [30].

The changes in the shape of H-G DQW potential according to the structure parameters as a function of the *z*-coordinate are given in Figure 1a–d, where $z_{0}$ is the asymmetry parameter. When the $z_{0}$-parameter is zero, the structure has a symmetrical character (Figure 1a). If $z_{0}\neq 0$, it becomes asymmetrical (Figure 1b–d). Thus, we will use the abbreviations H-G SDQW and H-G ADQW for the symmetric and asymmetric cases, respectively. The parameter $A_{1}$ causes a shift toward the higher energies in the confinement potential and a decrease in the effective width. The parameter $A_{2}$ causes an increase in the potential barrier height while the effective well width decreases. As seen in Figure 1d, for $A_{1}=0.5$, electrons on the third and fifth levels from the squared wave functions corresponding to the first six levels of the confined electron in H-G ADQW are located in the right well, and the others in the left well. For $A_{1}=0.2$, it is seen that electrons on the third and sixth levels from the squared wave functions corresponding to the first six levels of the confined electron in H-G ADQW are located in the right well (RW) and the others in the left well (LW). That is, electrons with energies of $E_{3}$ and $E_{5}$ in the first case and electrons with energies of $E_{3}$ and $E_{6}$ in the second case penetrate from LW to RW by tunneling.

For k=20 nm and $z_{0}=0$, the variation of the energies corresponding to the first six lower-lying levels of a confined electron within H-G SDQW as a function of the $A_{2}$-parameter for $A_{1}=0.2$ and $A_{1}=0.5$ are given in Figure 2a and Figure 2b, respectively. Solid (dashed) lines are for B=0 (B=15 T). As $A_{2}$ increases, the barrier height increases while the effective well width narrows, resulting in an increase in subband energies in the absence and presence of the magnetic field. The increase in the subband energies in the presence of the magnetic field is more pronounced since the magnetic field creates additional parabolic confinement. Without a magnetic field, the energy levels are two-folded and degenerate. First, the higher levels and then all energy levels begin to separate due to the increase caused in the energies by the magnetic field at small $A_{2}$ values ($A_{2}\approx 2$), and this behavior is observed at larger $A_{2}$ values ($A_{2}\approx 3$) as $A_{1}$ increases. Because the potential barrier for these energy levels is sufficiently thin, coupling between the wells increases. Since $A_{2}$ causes an increase in barrier width and a decrease in coupling between wells, a two-folded degeneration in the energies is observed again, even in the presence of the applied magnetic field at large $A_{2}$ values. Energy levels are two-folded degenerate due to the symmetry of the structure. The lower-lying bound states are two-folded degenerate since the barrier width is large to have no coupling between the wells. For higher bound states, the barrier thickness is narrower, and, therefore, the symmetric and antisymmetric states are separated due to the increasing energy with the magnetic field effect, and therefore degeneracy gradually disappears towards higher energy levels as in the fifth and sixth energy levels.

For a constant $A_{2}$-value ($A_{2}=2.0$), the variation of the energies that corresponds to the first six lower-lying levels of a confined electron within H-G ADQW as a function of the $z_{0}$-asymmetry parameter. Solid (dashed) lines are for $A_{1}=0.2$ ($A_{1}=0.5$) in the absence and presence of the magnetic field are given in Figure 3a and Figure 3b, respectively. While the first two energy levels are a direct decreasing function of the asymmetry parameter for both $A_{1}$ values, the first two energy levels are a direct decreasing function of the asymmetry parameter while the energies of the other levels increase and/or decrease according to the increasing $z_{0}$ parameter. The main reason for the oscillations of the energies is that the electrons at some other levels, except for the ground state, are localized in the right well. The level of electrons localized in the right well varies depending on the external parameters. For example, in Figure 3a, for $A_{1}=0.2$ and $z_{0}=0.15$, when there is no magnetic field, electrons in the third and fourth levels are localized in the right well, for A=0.5, at the same $z_{0}$ value electrons in the fourth and sixth levels are localized in the right well. In the presence of the magnetic field, for $A_{1}=0.5$ and $z_{0}=0.15$, all electrons are localized above the potential barrier, i.e., in a wider well. The localization in the right or left well of electrons at different levels varies depending on external parameters. Figure 4a,b have the same arrangements as Figure 3a,b, but these are for $A_{2}=4.0$. The variation of the energies according to the structure parameters and the applied external field is as in Figure 3a,b. The results in Figure 4 show the same trends and behaviors as those reported in Figure 3. However, in this case, a shift towards higher energies of all the reported states is observed, a situation that is in line with the displacement of the minimum of potential wells shown in Figure 1.

The variations of TACs and total refractive index (RIC) as a function of the incident photon energy corresponding to the (2–3) transition for $A_{2}=2.0$ (black lines) and (2–4) transition for $A_{2}=4.0$ (red lines) in H-G SDQW ($z_{0}=0$) with $A_{1}=0.5$ are given in Figure 5a and Figure 5b, respectively. Solid (dashed) lines are for B=0 (B=15T). For $A_{2}=2.0$, the (2–4) or (1–3) transition is forbidden, and for $A_{2}=4.0$, the (2–3) or (1–4) transitions are forbidden. This is because dipole matrix elements are zero due to the wave functions with the same parity. For $z_{0}=0$, the structure is symmetrical, and the diagonal matrix elements due to the even and odd characters of the wave functions are identical to zero ($M_{jj}=M_{ii}=0$). In addition, the dipole matrix elements of transitions for odd-to-odd or even-to-even (i.e., 1–3 or 2–4) quantum numbers disappear (meaning this kind of transitions are not allowed) since the envelope functions of these energy states have the same parity due to symmetry of the well. However, if the symmetry of the well is broken, the transitions mentioned become allowed.

To see more clearly how the TAC and RIC positions and amplitudes change concerning the structure parameters and magnetic field, the variation of the energy difference between related levels and the variation of RDME according to parameter-$A_{2}$ for only $A_{1}=0.5$ are given Figure 5c and Figure 5d, respectively. Here, black/red lines are for (2–3)/(2–4) transitions. Except for $E_{24}$ in the presence of the magnetic field and the values of $A_{2}\leq 2$, the difference between the indicated energy levels is usually an increasing function of parameter-$A_{2}$. After a certain *A* value, $E_{24}$ also begins to be an increasing energy function. In the range of $1\leq A_{2}\leq 2.5$, since the energy difference of $E_{23}=E_{3}-E_{2}$ ($E_{24}=E_{4}-E_{2}$) in the presence of a magnetic field is smaller (larger) than in the case without a magnetic field, both TAC and RIC positions shift to lower (higher) photon energies. In large $A_{2}$ values, since two-fold degeneracy starts in the energies in the presence of the magnetic field, $E_{23}$ becomes equal to $E_{24}$. Consequently, it is observed only one absorption peak exists, and so the peak positions of TAC and total RIC shift to high photon energies with the effect of the magnetic field.

In general, the positions of the absorption peaks depend on the transition energy between the two energy levels, while the change in peak amplitudes is attributed to the dipole matrix element. Let us examine the given equations for AC and RI according to the resonance conditions. Resonance conditions for incident photon energy are satisfied with the equality $\hbar \;\omega_{\max}=\sqrt{{\left(E_{ij}-\hbar \omega \right)}^{2}+{\left(\hbar \Gamma_{ij}\right)}^{2}}$ for which the linear AC has a maximum value, and so $\beta_{\max}^{\left(1\right)}\left(\omega \right)$, the maximum value of linear AC in Equation (Equation 4), becomes directly proportional to the energy difference and squared dipole matrix element in the form of $|M_{ij}{|}^{2}\left(E_{ij}+\sqrt{{\left(E_{ij}-\hbar \omega \right)}^{2}+{\left(\hbar \Gamma_{ij}\right)}^{2}}\right)$. The energy difference between the two energy levels is the dominant term on the peak positions of the ACs. The peak positions of ACs shift towards the higher (smaller) photon energies as the transition energy increases (decreases). By using the resonance condition in the form of $\hbar \omega_{\min}=\frac{1}{3}\left(E_{ij}+\sqrt{{\left(4E_{ij}\right)}^{2}+3{\left(\hbar \Gamma_{ij}\right)}^{2}}\right)$ for the incident photon energy that corresponds to the minimum value of third-order non-linear AC, it is seen that non-linear AC depend on the *I*-light intensity, dipole matrix element-$|M_{ij}{|}^{4}$, and transition energy-$E_{ij}$.

Furthermore, the positions and maximum and minimum values of the linear RIC, ${\left(\Delta n^{\left(1\right)}/n_{r}\right)}_{\max}$ and ${\left(\Delta n^{\left(1\right)}/n_{r}\right)}_{\min}$ for the resonance conditions $\hbar \omega_{\max(\min)}=E_{ij}\pm \hbar \Gamma_{ij}$, are proportional to $|M_{ij}{|}^{2}$ and $-|M_{ij}{|}^{2}$, respectively. Similarly, the positions and maximum and minimum values of third-order non-linear RIC, ${\left(\Delta n^{\left(3\right)}/n_{r}\right)}_{\max}$ and ${\left(\Delta n^{\left(3\right)}/n_{r}\right)}_{\min}$ for the resonance conditions $\hbar \omega_{\max\;\left(\min\right)}=E_{ij}\mp \frac{1}{\sqrt{3}}\hbar \Gamma_{ij}$, are proportional to $|M_{ij}{|}^{4}$ and $-|M_{ij}{|}^{4}$, respectively. Extreme values of linear and non-linear RICs are symmetrically positioned with respect to $\hbar \omega =E_{ij}$ [27]. In this context, it is seen that the positions and amplitudes of total ACs and RICs are consistent with the analyses made above about theirs and also the results of Figure 5c,d are consistent with these analyses.

For some transitions between the energy levels in H-G ADQW, which have parameters $z_{0}=0.10$, $A_{1}=0.2$, and $A_{2}=2.0$, the variation of TACs and RICs as a function of the incident photon energy, the variations of the energy difference between related levels and RDME according to parameter-$A_{2}$ are given in Figure 6a–d, respectively. Here black/red lines are for (1–3)/(2–4) transitions, and solid (dashed) lines are for B=0 (B=15T). For these parameters, all the energies considered are below the barrier and non-degenerate. Since the structure is asymmetrical, all possible transitions are allowed. In the absence of the magnetic field, the electrons in the first, third, and fifth levels are localized in the left well, and the electrons in the second, fourth, and sixth levels are localized in the right well. The (1–2), (2–3), and (3–4) transitions are not observed because the overlap integral are zero, while the (1–3) and (2–4) transitions are observed. With the effect of the magnetic field, the first, second, fourth, and sixth level electrons are localized in the left well, and the third and fifth level electrons are localized in the right well. In this case, the (1–3) transition is not observed since the overlap integral between the wave functions corresponding to the first and second levels is zero, but (1–2) and (2–4) transitions are observed. The peak positions of TAC and RIC corresponding to the transition of (2–4) shift towards the blue with increasing magnitudes appropriately with the results of Figure 6c,d. Furthermore, as seen in Figure 6d, the RDME for the (2–4) transition takes a substantial value under the magnetic field, a minimum in the total TAC occurs, and a small increase in the RIC is observed since the non-linear term becomes dominant.

For some transitions between the energy levels in H-G ADQW with $z_{0}=0.25$, $A_{1}\;=\;0.2$, and $A_{2}=2.0$, the variations of TAC and RIC as a function of the incident photon energy are given in Figure 7a and Figure 7b, respectively. Solid (dashed) lines are for B=0 (B=15T). As the asymmetry of the structure increases, the fourth and sixth level electrons are localized in the right well when there is no magnetic field, while the fifth level electron is completely localized in the right well, and the electron in the sixth level is more localized in the left well, although it is localized in both wells. The peaks of TACs and RICs corresponding to the (1–2) and (2–3) transitions shift towards the higher photon energies (blue shift), a minimum in the TAC occurs, and a small increase in the RIC for the (2–3) transition is observed since the non-linear term becomes dominant; this is more pronounced in the absence of a magnetic field.

To validate our study, in Figure 8, we present a comparison between the wave functions and energies for the ground state and the first excited state corresponding to an electron confined in a double quantum well with abrupt barriers, Figure 8a, and an electron confined in an H-G SDQW, Figure 8b. For the case of the H-G SDQW the confining potential is given by $V\left(z\right)=50{\left(z/k\right)}^{2}+228\;e^{-{\left(z/k\right)}^{2}}$ (in meV units). In the case of the rectangular double quantum well, we have taken a potential barrier whose height is 103 meV, which corresponds to a concentration of aluminum in the barriers of x=0.12. Notice that the bottom of the squared potential well is at the same energy as the bottom of the H-G SDQW. For the system of rectangular wells, the width of the central barrier is 6 nm while the width of each of the two symmetric wells is 5.1 nm. The transition energy between the ground state and the first excited state in the H-G SDQW system is 9.9 meV, while for the double rectangular well it is 10.3 meV. The agreement between wave functions, energies of the two reported states, and the transition energy for the two systems is evident. Among the advantages of using the H-G SDQW model to simulate a double quantum well structure lies in the fact that, in this case, non-abrupt variations of aluminum concentration at the interfaces from the well region to the barrier region can be considered, a situation that is in excellent agreement with the interdiffusion phenomena in low-dimensional semiconductor heterostructures. Additionally, the H-G SDQW system allows the introduction of a non-abrupt dependence on the effective mass, adapted to the aluminum concentration’s functional variation.

## 4. Conclusions

Using the effective mass and parabolic conduction band approximations, in this paper, we report the optical absorption and refractive index coefficients in symmetric and asymmetric double quantum wells. The one-dimensional confinement potential has been modeled by the sum of an internal Gaussian barrier and a harmonic potential under the effects of an externally applied magnetic field. The solution of the eigenvalues differential equation has been obtained via a diagonalization method considering a base of sine-like orthonormal function. To calculate the linear and third-order non-linear optical absorption and refractive index coefficients, a two-level approach is used in the density matrix expansion. Among the main findings of this research, we can report the following: (*i*) the depth of the two potential wells and the degree of coupling between them can be controlled by variations in the $A_{1}$ and $A_{2}$ structural parameters; (*ii*) the $z_{0}$ asymmetry parameter is useful to simulate effects of electric fields and, thus, manipulate the selection rules of dipole moments, giving rise to new transitions that are optically prohibited in symmetric heterostructures; (*iii*) the variations in the asymmetry parameter of the heterostructures, $z_{0}$, generate oscillations in the functional dependence with $z_{0}$ of the confined electronic states, a situation that becomes more noticeable for highly excited states; (*iv*) depending on the parameters that control the double quantum well system, the applied magnetic field may be responsible for shifts to red or blue of the different transitions considered in the optical properties studied; and, finally, (*v*) the presence of bleaching in the absorption coefficient for certain geometries evidences the limitations of the model used to study the optical properties in these systems.

Finally, we want to say that the double spatial confinement model is helpful to describe the physics of coupled dot-ring systems considering the rotation of the potential in Equation (Equation 3) around the x-axis, for example, and taking as a reference point the minimum to the left of the potentials shown in Figure 1. Likewise, this model can be extended to the study of colloidal spherical quantum dots such as core/shell structures. Research in this regard is under development and will be published on another occasion.

## Figures and Tables

**Figure 1 nanomaterials-13-00892-f001:**
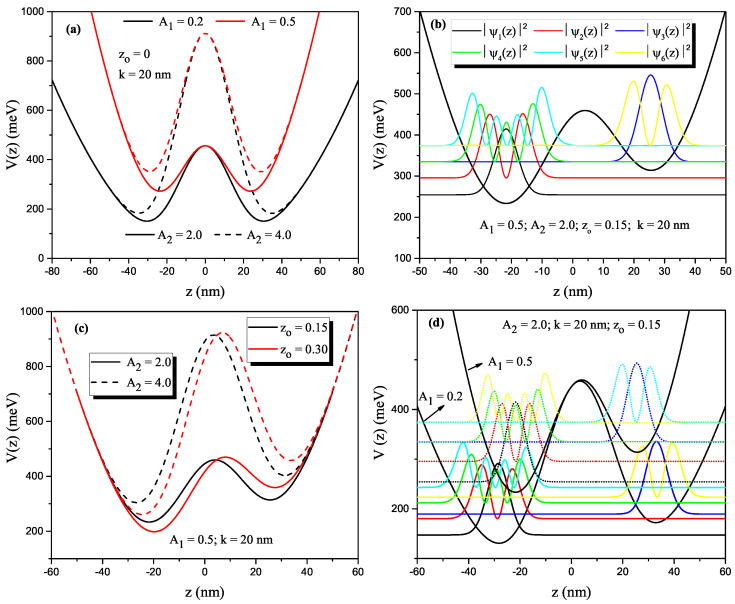
Harmonic-Gaussian DQW confinement potential profile for a constant value of k=20 nm versus the *z*-growth direction coordinate. Harmonic-Gaussian symmetric DQW, solid (dashed) lines are for $A_{2}=2.0$ ($A_{2}=4.0$) and black (red) lines $A_{1}=0.2$ ($A_{1}=0.5$) (**a**). For $z_{0}=0.15$, $A_{1}=0.5$, and $A_{2}=2.0$ harmonic-Gaussian asymmetric DQW confinement profile and squared wave functions corresponding to the first six energy levels (**b**). For different $z_{0}$-values and some values of the structure parameters, the harmonic-Gaussian asymmetric DQW profile (**c**), and harmonic-Gaussian asymmetric DQW confinement profile and squared wave-functions corresponding to the first six energy levels for the constant values of $A_{2}$ and $z_{0}$ but two different values of the parameter-$A_{1}$ (**d**).

**Figure 2 nanomaterials-13-00892-f002:**
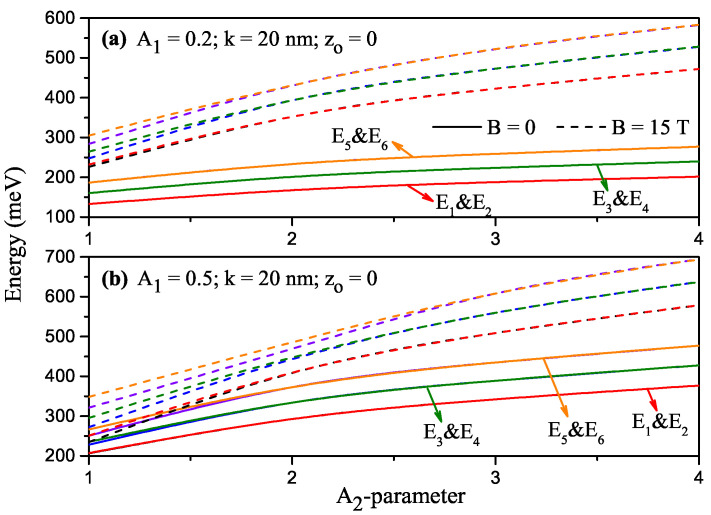
For k=20nm and $z_{0}=0$, the variation of the energies corresponding to the first six lower-lying levels of a confined electron within Harmonic-Gaussian symmetric DQW as a function of the $A_{2}$-parameter: $A_{1}=0.2$ (**a**) and $A_{1}=0.5$ (**b**). Solid (dashed) lines are for B=0 (B=15T).

**Figure 3 nanomaterials-13-00892-f003:**
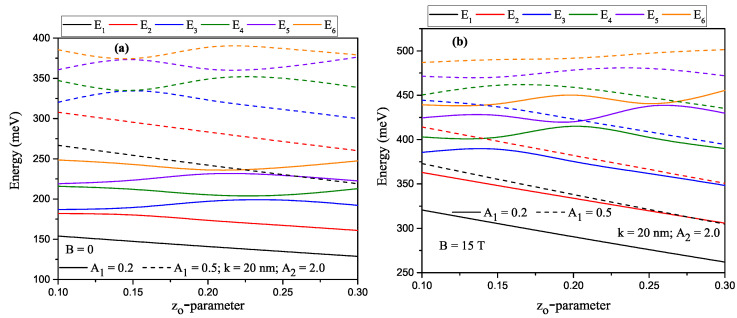
For $A_{2}=2.0$, the variation of the energies that corresponds to the first six lower-lying levels of a confined electron within Harmonic-Gaussian asymmetric DQW as a function of the $z_{0}$-parameter. Solid (dashed) lines are for $A_{1}=0.2$ ($A_{1}=0.5$). Results are for B=0 (**a**) and B=15T (**b**).

**Figure 4 nanomaterials-13-00892-f004:**
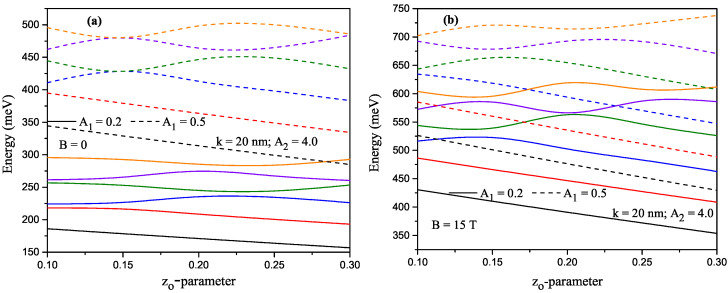
Results are as in Figure 3, but for $A_{2}=4.0$.

**Figure 5 nanomaterials-13-00892-f005:**
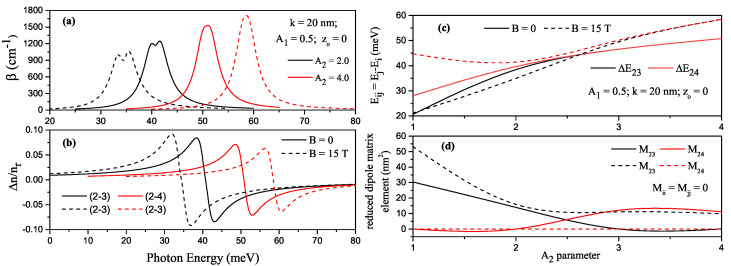
For some transitions between the energy levels in harmonic-Gaussian symmetric DQW ($z_{0}=0$) with $A_{1}=0.5$, the variation of total absorption coefficients as a function of the incident photon energy (**a**) and the variation of total refractive index as a function of the incident photon energy (**b**). Here, black (red) lines are for $A_{2}=2.0$ ($A_{2}=4.0$), according to parameter-$A_{2}$. The variation of the energy difference between related levels (**c**) and the variation of reduced dipole matrix element (**d**), where black/red lines are for (2–3)/(2–4) transitions. Solid (dashed) lines are for B=0 (B=15T).

**Figure 6 nanomaterials-13-00892-f006:**
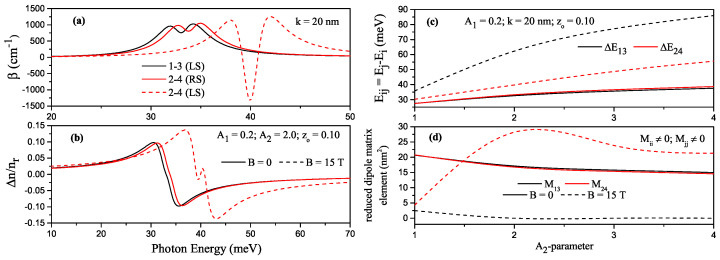
For some transitions between the energy levels in harmonic-Gaussian asymmetric DQW with $z_{0}=0.10$, $A_{1}=0.2$, and $A_{2}=2.0$, the variation of total absorption coefficients as a function of the incident photon energy (**a**), the variation of total refractive index as a function of the incident photon energy (**b**). According to the parameter-$A_{2}$, the variation of the energy difference between related levels (**c**) and the variation of reduced dipole matrix element (**d**), where black/red lines are for (1–3)/(2–4) transitions. Solid (dashed) lines are for B=0 (B=15T).

**Figure 7 nanomaterials-13-00892-f007:**
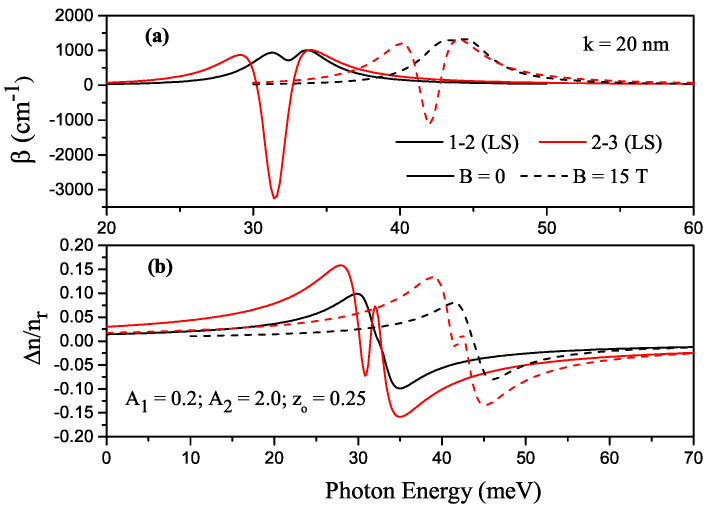
For some transitions between the energy levels in Harmonic-Gaussian asymmetric DQW with $z_{0}=0.25$, $A_{1}=0.2$, and $A_{2}=2.0$, the variation of total absorption coefficients as a function of the incident photon energy (**a**) and the variation of total refractive index as a function of the incident photon energy (**b**). Solid (dashed) lines are for B=0 (B=15T).

**Figure 8 nanomaterials-13-00892-f008:**
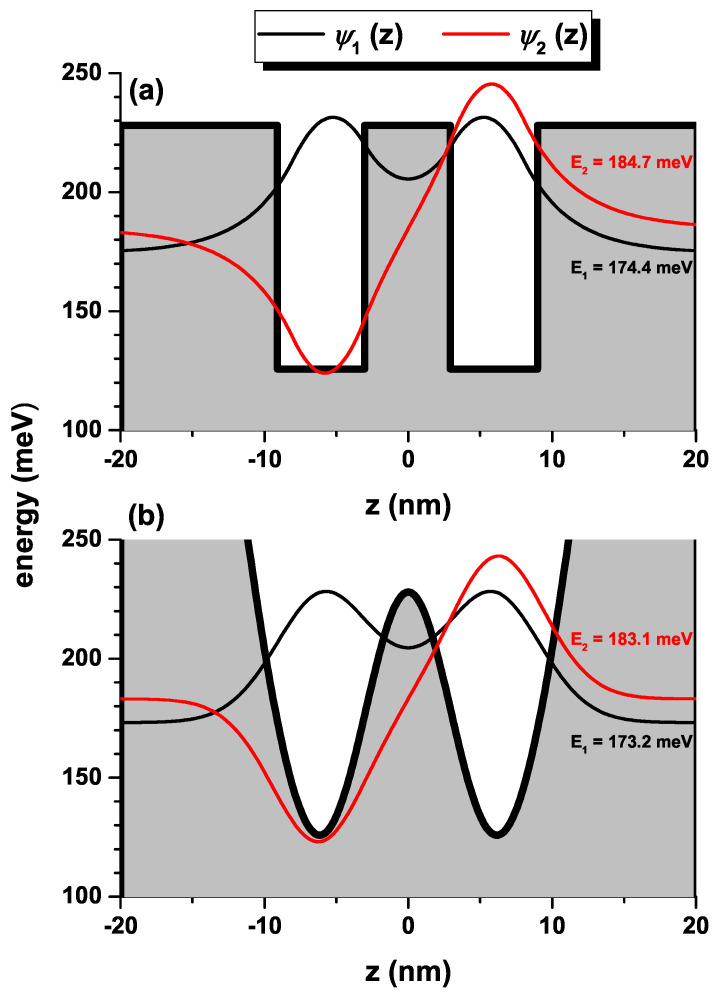
The ground and the first excited states wave functions for a confined electron in rectangular shaped (**a**) and harmonic-Gaussian (**b**) symmetric double quantum wells. The corresponding energies are also depicted.

## Data Availability

No new data were created or analyzed in this study. Data sharing is not applicable to this article.

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
