# Peer review of "Harmonic-Gaussian Symmetric and Asymmetric Double Quantum Wells: Magnetic Field Effects"

_nanomaterials, 2023, doi:10.3390/nano13050892_

Round 1

Reviewer 1 Report

Esin Kasapoglu et al. studied the linear and nonlinear optical properties of an electron in double quantum wells by using effective mass and parabolic band approximations. They calculated the eigenvalues, eigenfunctions, and absorption/refraction coefficients in the quantum wells. The methods reported are solid, but I am not convinced that they have broad audience on this journal. The models seem over-simplified for material research, and there is no clear clue how these model can guide experimental studies. Though authors claim some applicable cases, detailed discussion and analysis are missing. Authors are recommended to take further study with experimental supports.

Author Response

Referee 1

The Referee:

Esin Kasapoglu et al. studied the linear and nonlinear optical properties of an electron in double quantum wells by using effective mass and parabolic band approximations. They calculated the eigenvalues, eigenfunctions, and absorption/refraction coefficients in the quantum wells. The methods reported are solid, but I am not convinced that they have broad audience on this journal. The models seem over-simplified for material research, and there is no clear clue how these models can guide experimental studies. Though authors claim some applicable cases, detailed discussion and analysis are missing. Authors are recommended to take further study with experimental supports

Our reply:

We want to thank and express our gratitude to the Referee for his/her excellent report, which we believe has helped us substantially improve the quality and clarity of our manuscript.

About the Referee's opinion, we want to say that in the present study we have resorted to a confinement potential that has proven to be useful to describe a molecular system in which inversion symmetry occurs and we have adapted it to describe the physics of symmetric and asymmetric double quantum wells. In the case of the asymmetric double quantum wells, we have observed that this system allows us to simulate electric field effects in double quantum wells. In the case of these systems it is possible to introduce a change in the interband transition associated with spatially direct excitons towards spatially indirect excitons. The utility of these systems is that it is highly feasible to increase the lifetime for excitonic recombination by several orders of magnitude. Obviously, that is not the purpose of this study and for this reason extending our research towards the calculation of excitonic states in double quantum wells is beyond the scope of our article.

Another of the advantages of the model used in this article lies in the fact that a potential that varies smoothly along the growth coordinate of the structure makes it possible to simulate interdiffusion effects that are present at the well-barrier interfaces of the low dimensional heterostructures.

Anyway, considering the excellent suggestion of the Referee, we have decided to validate our model and we have implemented a comparison between the first two confined states in a double quantum well with abrupt barriers and an H-G SDQW. We have added these results in the revised version of the manuscript in Fig. 8 together with the corresponding discussion.

In the Conclusions section we have added the following comment:

Finally, we want to say that the double spatial confinement model is helpful to describe the physics of coupled dot-ring systems considering the rotation of the potential in Eq. (3) around the x-axis, for example, and taking as a reference point the minimum to the left of the potentials shown in Fig. 1. Likewise, this model can be extended to the study of colloidal spherical quantum dots such as core/shell structures. Research in this regard is under development and will be published on another occasion.

We hope that our answers and comments are satisfactory and that the Referee considers that our article is suitable in its present form to be published in the Nanomaterials journal.

Reviewer 2 Report

The authors studied the symmetic and asymmetric double quantum wells under magnetic field applied to study the inversion of Ammonia. The results presented here are interesting and deserve publishing but with a few improvements.

1) Except for the references related to those mentioned in the text, the PT potential and others can also be used to study diatomic molecules. For example, Int. J. Quan. Chem. 86, 265 (2002); J. Phys. A: Math. Theor. 42, 035303 (2009), Advances in High Energy Physics, Article ID 9105825, 7 pages (2018);etc. I suggest the authors recheck recent advances in this field, in particular those double well potential related to hyperbolic potentials whose soultions are expressed by Heun functions, Journal of Mathematical Chemistry 60 (4), 605-612 (2022), etc.

Author Response

Referee 2

The Referee:

The authors studied the symmetic and asymmetric double quantum wells under magnetic field applied to study the inversion of Ammonia. The results presented here are interesting and deserve publishing but with a few improvements.

Our reply:

We want to thank and express our gratitude to the Referee for his/her excellent report, which we believe has helped us substantially improve the quality and clarity of our manuscript.

The Referee:

1) Except for the references related to those mentioned in the text, the PT potential and others can also be used to study diatomic molecules. For example, Int. J. Quan. Chem. 86, 265 (2002); J. Phys. A: Math. Theor. 42, 035303 (2009), Advances in High Energy Physics, Article ID 9105825, 7 pages (2018); etc. I suggest the authors recheck recent advances in this field, in particular those double well potential related to hyperbolic potentials whose soultions are expressed by Heun functions, Journal of Mathematical Chemistry 60 (4), 605-612 (2022), etc.

Our reply:

We want to thank the Referee for the suggested references, which are completely relevant for the construction of the state of the art of our article. These references will be very useful for the development of our future research.

At the end of the second paragraph of the introduction section, we added the follwoing text (the corresponding References have been included in the References section):

Dong and Lemus reported the ladder operators for the modified Pöschl-Teller potential \cite{Ref1}. Particularly, they found a closed form of the normalization constants of the wave function by using two different methods and calculated analytical expressions for the matrix elements derived from the ladder operators. Using the exact quantization rule, Gu \textit{et al.} calculated the energy spectra for modified Rosen–Morse potential \cite{Ref2}. In the same way, Dong \textit{et al.} reported semiexact Solutions of the Razavy Potential \cite{Ref3}. In their work, they show how to find the wave function exact solutions, which are given by the confluent Heun functions. Additionally, their method has been extended to the calculations of the asymmetric double well potential \cite{Ref4}.

We hope that our answers and comments are satisfactory and that the Referee considers that our article is suitable in its present form to be published in the Nanomaterials journal.

Reviewer 3 Report

The work uses double quantum well wavefunction to simulate the NH3 molecule inversion. The work is interesting, however, there are still some questions that need to be answered.

1. It is better to provide the schematic image for the relation between the ammonia inversion and double quantum well potential.

2. How does the author get the parameter for numerical calculations? Is there any reference supporting these values?

3. Is there any experimental result related to this theoretical work?  Do they match?

Author Response

Referee 3

The Referee:

The work uses double quantum well wavefunction to simulate the NH3 molecule inversion. The work is interesting, however, there are still some questions that need to be answered.

Our reply:

We want to thank and express our gratitude to the Referee for his/her excellent report, which we believe has helped us substantially improve the quality and clarity of our manuscript.

The Referee:

  1. It is better to provide the schematic image for the relation between the ammonia inversion and double quantum well potential.

Our reply:

We want to apologize to the Referee about an incorrect presentation that we have made about the scope of our article. The purpose of our work is to use the molecular potential of the NH3 molecule, to implement it in the study of low-dimensional systems of the type of a double quantum well. This type of molecular potentials extended to low-dimensional systems can be applied without much difficulty to structures such as coupled quantum rings, dots and quantum rings coupled together, core/shell type quantum threads, core/shell colloidal quantum dots. shell, among many others. To prevent us from generating an incorrect expectation about our article, we have decided to change e. qualification. In the preliminary version the title of the article was “Harmonic-Gaussian symmetric and asymmetric double quantum wells under applied magnetic field: Potential Function for the Inversion of Ammonia”. In the modified version of our article, the title reads as follows: “Harmonic-Gaussian symmetric and asymmetric double quantum wells: magnetic field effects”.

The Referee:

  1. How does the author get the parameter for numerical calculations? Is there any reference supporting these values?

Our reply:

In the first paragraph of the Results and Discussion section we have placed the parameters used for the effective mass, the dielectric constant, the relaxation times, the height of the potential barrier, the electron density, and the intensity of the resonant incident radiation that excites intersubband transitions. These parameters have been taken from Reference [30] of the revised version of our manuscript and are typical values for GaAs surrounded by AlxGa1-xAs. Using the values of effective mass and dielectric constant reported here, the effective Bohr radius and effective Rydberg energy can be calculated, which are 10 nm and 5.7 meV, respectively. From the four panels of Fig. 1, it can be seen that the average value of the distance between the two minima of the two potentials shown is of the order of 20-30 nm, which corresponds to a range of 2 to 3 effective Bohr radii. Also, the height of the central barrier is of the order of 200 meV, which is in the range of 35 effective Rydbergs. In this way, it can be concluded that the dimensions of the structures and the heights of the potential barriers allow the effects of quantum confinement and guarantee the coupling of the two potential wells. This situation is key to be able to modify the optoelectronic properties of devices based on double quantum wells. The parameters A1, A2 and z0 have then been chosen in such a way that the semiconductor heterostructures to be simulated have dimensions of the order of the effective Bohr radius with potential barriers that fit the band offset of the GaAs and AlxGa1-xAs semiconductors with concentrations of aluminum of the order of x=0.1 to x=0.3

The Referee:

  1. Is there any experimental result related to this theoretical work? Do they match?

Our reply:

About the Referee's questions, we want to say that in the present study we have resorted to a confinement potential that has proven to be useful to describe a molecular system in which inversion symmetry occurs and we have adapted it to describe the physics of symmetric and asymmetric double quantum wells. In the case of the asymmetric double quantum wells, we have observed that this system allows us to simulate electric field effects in double quantum wells. In the case of these systems it is possible to introduce a change in the interband transition associated with spatially direct excitons towards spatially indirect excitons. The utility of these systems is that it is highly feasible to increase the lifetime for excitonic recombination by several orders of magnitude. Obviously, that is not the purpose of this study and for this reason extending our research towards the calculation of excitonic states in double quantum wells is beyond the scope of our article.

Another of the advantages of the model used in this article lies in the fact that a potential that varies smoothly along the growth coordinate of the structure makes it possible to simulate interdiffusion effects that are present at the well-barrier interfaces of the low dimensional heterostructures.

Anyway, considering the excellent questions of the Referee, we have decided to validate our model and we have implemented a comparison between the first two confined states in a double quantum well with abrupt barriers and an H-G SDQW. We have added these results in the revised version of the manuscript in Fig. 8 together with the corresponding discussion.

In the Conclusions section we have added the following comment:

Finally, we want to say that the double spatial confinement model is helpful to describe the physics of coupled dot-ring systems considering the rotation of the potential in Eq. (3) around the x-axis, for example, and taking as a reference point the minimum to the left of the potentials shown in Fig. 1. Likewise, this model can be extended to the study of colloidal spherical quantum dots such as core/shell structures. Research in this regard is under development and will be published on another occasion.

We hope that our answers and comments are satisfactory and that the Referee considers that our article is suitable in its present form to be published in the Nanomaterials journal.

Round 2

Reviewer 2 Report

The authors considered those points made by the reviewer.